# Critical Review on the Different Roles of Exosomes in TNBC and Exosomal-Mediated Delivery of microRNA/siRNA/lncRNA and Drug Targeting Signalling Pathways in Triple-Negative Breast Cancer

**DOI:** 10.3390/molecules28041802

**Published:** 2023-02-14

**Authors:** Manosi Banerjee, Vijayarangan Devi Rajeswari

**Affiliations:** School of Bioscience and Technology, Vellore Institute of Technology, Vellore 632014, Tamil Nadu, India

**Keywords:** triple-negative breast cancer, exosomes, biomarkers, therapeutics, signalling, drug-resistant

## Abstract

Triple-negative breast cancer is the most potent metastatic type of breast cancer that can spread to other body parts. Chemotherapy and surgical intervention are the sole treatments for TNBC, owing to the scarcity of therapeutic targets. Manipulation of the membranes as per the desired targets of exosomes has recently gained much attention as a drug delivery method. Despite their known roles in different diseases, very few studies have focused on signalling that triggers the metastasis of triple-negative breast cancer to other body parts by exosomes. This article highlights the significant roles of exosomes associated with TNBC, the involvement of exosomes in breast cancer diagnosis, progression, and the treatment of triple-negative breast cancer by the exosomes as a drug delivery system. This review paper also illustrates the role of exosomes in initiating EMT in breast cancer, including novel signalling.

## 1. Introduction

### 1.1. Triple Negative Breast Cancer

Breast cancer is the leading cancer in women, accounting for 27% of all cancers in India [1]. TNBC subtypes account for 15–20% of breast cancer [2]. Mutation in BRCA1/2 causes breast cancer. TNBC is aggressive and eventually transfers to other body organs such as the lungs, brain, heart, bone, and liver [3]. The survival rate (5 years) for triple-negative breast cancer patients is low (77% in contrast to 93% for other breast cancer subtypes) [4]. The receptor’s expression profile in TNBC is very complicated and contradictory to what its name suggests. The lack of phenotypic receptors, such as estrogen and progesterone, reduces the expression of human epidermal growth factor 2. Molecular subtypes of triple-negative breast cancer are (i) basal-like 1, (ii) basal-like 2, (iii) immunomodulatory, (iv) mesenchymal, (v) mesenchymal stem-like, and (vi) luminal androgen receptor [5,6]. The prediction of a 5-year survival rate can be more accurate depending on the subtyping of TNBC. The current statistics show that it differs in basal-like tumours (68%) in contrast to non-basal-like tumours (79%). The effect of chemotherapy is different in different subtypes of breast cancer [7]. 

Clinically, treatment for TNBC is very restricted due to the lack of specific targets [8]. The dearth of effective targeted delivery leads to poor prognosis and treatment outcomes [9]. Hence, side effects are associated with conventional chemotherapy, which can simultaneously be fatal and distressing, but it is the sole option available [10]. Chemotherapeutics have a small therapeutic window when used together [11]. Furthermore, most of them cannot penetrate the blood–brain barrier (BBB), limiting therapeutic options for brain tumours, particularly those caused by metastasis of TNBC [12,13]. Regrettably, even when the first line of therapy is there, the recurrent rate is much higher and occurs within three years.

### 1.2. Need of Delivery Vehicle—Exosomes as an Answer

Restrictions on anticancer drug potency are due to poor drug solubility, half-life, and a drug-resistant result of inadequate binding of the drugs with the specific ligands of the tumours [14]. To find a proper solution to all these clinical challenges, testing of nano-formulations such as hydrogels, microneedle patches, biodegradable polymers, microspheres, and nanoparticles has been under research [15]. The ability of the nano-formulations was proved in pre-clinical trials, but off-target effects and toxicity have disrupted their role in the clinical trials [16]. All challenges described above justify the need for new and improved nanocarriers with adequate properties such as low toxicity, longer half-life, and biocompatibility. Developing engineered nanocarriers can be a solution that can target specific ligands [17]. To fulfil these demands, scientists have devised extracellular vesicles, i.e., exosomes, as the delivery system for various multiple chemotherapeutics. The toxicity of other nano-formulations makes exosomes efficient delivery agents with a small size and more biocompatibility. Exosomes have better pharmacokinetics and pharmacodynamics compared to other nano-delivery systems [18]. The exosomes can even cross the blood barrier and can target tumours in the brain. Hence, when TNBC metastasizes to the brain, exosomes can quickly cross the BBB and target it [19]. The exosomes are derived from different biological sources and act not too similarly in vivo. The cellular origins of exosomes are significant [20]. Drug loading capacity also depends on the source from which exosomes are extracted. Extracted exosomes from macrophages show higher loading efficiency than those derived from pancreatic cells [21,22]. Exosomes separated from donor cells that are not the same as the recipient cells are more effective as delivery vehicles. Exosomes isolated from particular cell types exhibit organotropism toward various cancers, accumulating mostly in the organ’s primary cell type rather than in adjoining cell types [23].

## 2. Extracellular Vesicles

Intercellular communication in all cells results from the extracellular vesicles produced by almost all cell types. EVs can be divided into three categories, i.e., microvesicles (MVs), exosomes, and apoptotic bodies, depending upon the size and the generation of pathways [24]. Among them, exosomes regulate the normal functioning of cells. They are also involved in the pathology of several diseases, including neurodegenerative and autoimmune, cardiovascular, and cancer [25,26,27,28]. These features have drawn much attention to exosomes; therefore, they have been widely researched in cancer biology and therapy. The size of exosomes is 30–150 nm, synthesized by the endocytic process and produced by the fusion of multi-vesicular bodies (MVBs) with the plasma membrane [29].

Exosomes are overexpressed with distinguished membrane proteins called tetraspanins and proteins such as CD9, CD37, CD63, CD81, and CD82. Tumour susceptibility gene (TSG)-101 and Alix of the exosome sorting complex required for transport (ESCRT) are some of the exosome’s biomarkers [30]. The ever-increasing literature suggests that exosomes regulate the normal functioning of cells. The exosome components (lipids, nucleic acids, metabolites, proteins, and peptides) reflect their cellular origin. Various databases have all the detailed compositions of exosomes to characterize them properly, such as ExoCarta, Vesiclepedia, and EV miRNA [31,32,33].

## 3. Exosomes in TNBC

### 3.1. Exosomes Carry Signature Markers from Their Cells of Origin

The number of exosomes released by TNBC (triple-negative breast cancer) cells is more incredible than more significant normal healthy cells. The augmentation of TSAP6, a p53-regulated gene product that governs the exosomal secretion signalling transcription by activating p53, which is frequently aberrantly driven in cancer cells, could alter the rise in exosome shedding from cancer cells. The most associated biomarkers with exosomes are tetraspanins (CD9, CD63, CD81), proteins of the endosome system (TSG101, Rab-GTPase), and heat shock chaperones (HSP70, HSP90). These are all the characterizations of all exosomes [34,35,36].

Exosomes produced by cancer cells have different content than normal cells. The microRNA content in the exosomes extracted from cancer cells has different signature microRNA and acts similar to “fingerprints”, which can identify their origin. Hence, they can also serve as a biomarker. A protein senescence-associated secretory phenotype (SASP) that can enhance tumour growth is one of the constituents of exosomes [37].

Chemotherapeutics induce apoptosis. Some cancer cells experience therapeutic-induced senescence (TIS), which enables them to stay metabolically active but lose their capacity to metastasize. This condition makes them more resistant to chemotherapies. Kavanagh et al. found that exosomes collected from TIS TNBC CAL51 cells had much larger quantities than non-senescent cells. Concerning non-senescent TNBC cells, the researchers found a remarkable increase in the amounts of 142 proteins in exosomes from these TIS cells. Essential proteins, which are all involved in (i) cell proliferation, (ii) ATP depletion, (iii) apoptosis, and (iv) SASP factors, are more prevalent in exosomes from TIS cells, implying that the abolition of these proteins from TIS cells through exosomes permits cancer senescent cells to stay feasible [38].

Exosomes also play an important role as a messenger and can communicate between the recipient cells and donor cells through the microRNA, which they carry, which becomes translated into proteins. Communication can involve immune suppression, microenvironment changes, angiogenesis, immune escape and invasion, and tumour progression and repression.

### 3.2. Exosomes for Diagnosis of Triple-Negative Breast Cancer

With the advancement of nano-theragnostics, exosome profiling, diagnosis, and potential disease therapy are possible. A ‘liquid biopsy’ enables the study of EVs and exosomes. It also aids in analysing circulating DNA from tumours and cells from tumours. The exosomes carry specific biomarkers that help to diagnose triple-negative breast cancer. The content of the exosomes is different for patients with triple-negative breast cancer [39]. Proteins that act as biomarkers for TNBC are carcinoembryonic antigen (CEA), survivin, and CA 15-3(cancer antigen 15-3) [40]. The levels of these biomarkers are elevated in patients suffering from TNBC. The microRNA present inside the exosomes also plays a vital role in diagnosing. A specific microRNA identified as a biomarker is miR-373, which is elevated in TNBC patients and further downregulates the expression of estrogen [41]. The Wnt signalling is responsible for metastasis and chemoresistance in TNBC. Myotubularin-related protein 3 is downregulated by miR-1910-3p, which also stimulates the NF-κB and wnt/β-catenin signalling pathways and, in turn, accelerates the development of breast cancer. When combined with the conventional tumour marker CA153, serum miR-1910-3p in exosomes is a potent diagnostic sign that increases the sensitivity of breast cancer detection. In conclusion, miR-1910-3p in serum exosomes may be a new molecular indicator for the detection of breast cancer. Liquid biopsy investigates intra-tumour heterogenicity and the tumour microenvironment, while single-tissue biopsy does not include the microenvironment [42,43] Figure 1.

### 3.3. Exosomes Initiating the Epithelial-Mesenchymal Transition in Breast Cancer and TNBC Influencing the Metastasis to Other Organs

EMT is a process through which normal cells transfer to mesenchymal cells. Mesenchymal cells have the property of invasion and malignancy, leading to the worst prognosis and TNBC migrating to other organs. The EMT transcription factors involved are β-catenin, TWIST, zinc finger protein, SNAIL2 (zinc finger protein SNAI 1), SLUG (SNAI 2 gene), ZEB1 (zinc finger E-box-binding homeobox 1), and ZEB 2. Studies have proven that altering these transcription factors is not enough to cease the epithelial–mesenchymal transition. The exosomes are involved in intercommunication between the cancer cells. The exosomes have various contents, which can regulate the transcription involved in the epithelial-to-mesenchymal transition [44]. The literature, with various reports, shows that exosomes are involved in the epithelial–mesenchymal transition in breast cancer.

Synthesis of exosomes takes place when intercommunication between notch receptors, ligands (JAGs) (jagged protein), and regulators (ADAM 10/17) (A disintegrin and metalloproteinase) occurs. ASPH (aspartate beta-hydroxylase) initiates the Notch cascade to synthesis or release of pro-metastatic exosomes. Normal breast cells have muted ASPH. The activation of ASPH by the NOTCH signalling gives rise to aggressive tumour progression in breast cancer with the help of exosomes [45]. Hippo signalling pathways are essential in restricting organ size. They inhibit oncogenic co-activators such as YAP (yes-associated protein) [46]. Inactivation of the Hippo pathway initiates the TEA (transcriptional enhancer factor) domain transcription factor to bind with the unphosphorylated YAP/TAZ (tafazzin), which translocates it to the nucleus [47]. TEAD (transcriptional enhancer factor domain TEF1)-influences the target genes. The transcription enhances the expression of mesenchymal markers such as Vimentin and N-cadherin and suppresses the expression of epithelial markers including E-cadherin [48]. The YAP gene cross-communicates with transcription factors such as ZEB1, SNAIL, and SLUG, which initiates epithelial–mesenchymal transition in cancer cells. Exosomes from mesenchymal-stem-cell-derived adipocytes enhance Hippo-induced epithelial–mesenchymal transition in breast cancer cells [49].

The Wnt/β-catenin signalling pathway is involved in the epithelial–mesenchymal transition. Specifically, a β-catenin molecule is involved in the epithelial–mesenchymal transition [50,51]. Scientists reported that exosomes containing Wnt ligands could influence Wnt signalling, which is responsible for EMT synthesis. Macrophage-derived exosomes containing Wnt5a initiate the Wnt/β-catenin signalling pathway in breast cancer cells [52]. Exosomal Wnt7a is accountable for the metastasis of breast tumours to the lungs [53]. In human breast fibroblasts, the exosome-mediated release of miR-9 produces cancer-associated fibroblast-like characteristics. Hence, regulating the Wnt ligands encapsulated in exosomes can be reasonable therapeutic control for cancer cells. Modulation of Wnt signalling by proteins derived from exosomes can inhibit metastasis [54].

The lncRNAs MALAT1 (metastasis-associated lung adenocarcinoma transcript 1) and ID4 (inhibitor of DNA-binding 4) cause an upregulation of the circular RNA circ 0076611 in exosomes released by TNBC. This circular RNA circ 0076611, released from the exosomes of TNBC cells, interacts with MYC (proto-oncogenes) and VEGFA (vascular endothelial growth factor A) mRNAs, further initiating cell proliferation and migration of TNBC cells [55]. Exosomes secreted by the TNBC cell line containing MMP-1 (Matrix Metallopeptidase 1) interact with the PAR1( protenase-activated receptor 1), a G protein receptor that initiates the EMT, which enables the breast cancer tumour to metastasize to the lungs [56]. miR-939 in exosomes downregulates the expression of VE-cadherin, destroys the barrier function of endothelial monolayers, and initiates tumourigenesis in breast cancer [57]. miR-155 is upregulated in TNBC, induces epithelial–mesenchymal transition, and plays an essential role in resistance [58,59] Figure 2.

## 4. Exosomes—Production, Preservation, Loading, Surface Engineering, and Exosome Hybrids in Drug Delivery

The production extraction of exosomes from the cells includes ultrafiltration, ultracentrifugation and precipitation, and immunoaffinity [60]. All these processes often have low-quality yield with contamination. Acoustic and fluidic device-based purification methods are often used to improve the quality [60,61]. Tangential flow filtration is often used to extract exosomes from the cell lines, improving good quality yield. Laboratory and industrial-grade bioreactors are also used for large-scale productions [62]. On the other hand, these bioreactor-based approaches involve inherent procedures that alter and modify the phenotypics of the cells over time, lowering the exosome quality. When T cells are agitated at 180 rpm, interleukin (IL)-2 levels in their exosomes are lowered; as a result, it is critical to carefully identify and develop a process for huge-scale exosome manufacturing that ensures exosome quality without negatively affecting the exosome source [63]. External stimuli such as chemical and electrical stimulation, hypoxia conditions, and silencing of specific endo lysosomal traffic proteins have shown better exosome yield from the cells [64,65]. 

Preservation of the exosomes often involves storing them at −80 °C or 4 °C. Keeping them at −80 degrees Celsius shows perfect enzymatic activity of beta-glucuronidase [66]. Exosome aggregation during the lyophilization step is reduced in the B16BL6 cell line when exosomes are stored using cryoprotectants such as trehalose [67]. Contrasting results were seen by Akers et al., that lyophilization leads to a reduction in the number of exosomes and RNA content compared to when it is stored at room temperature [68]. Modifying the exosomes with the arginine, the cell retained its quality after lyophilization. Continuous freezing and thawing result in a decrease in the number of exosomes [69].

Loading the exosomes with the desired compound is important. The exosomes can be used as a delivery agent for molecular drugs such as microRNA and siRNA, which are digested by the nuclease present in the body [70]. Other biomolecules include nucleic acids, lipids, proteins, and chemotherapeutic drugs. At present, there are two methods of loading: (a) endogenous and (b) exogenous methods. In the endogenous methods, the exosomes already have the compound of interest with the content of the protein or nucleic acid pre-loaded by the overexpression of the protein of interest; they are secreted [71].

In the exogenous methods, the exosomes are loaded by the interest of protein or nucleic acids with the help of artificial processes such as electroporation, co-incubation, sonication, lipofection, pH gradient, and calcium chloride methods [72]. All loading methods have cons and pros associated with them. The type of molecule of interest is vital for loading. For example, the loading of mRNAs into exosomes is often done by endogenous methods by enhancing the overexpression by transfecting the cell with plasmid DNA to trigger the production of desired biomolecules, which an electric signal can further stimulate. This method is known as the nano-proration method, which increases the yield of exosomes by 50-fold and 1000-fold in the exosomes having mRNAs compared with the standard methods [70]. The loading of microRNA and siRNA is performed through exogenous methods. One of the methods of sonication was seen to be effective in the case of breast cancer, where siRNA against HER2 was loaded, and a further reduction in the HER2 (human epidermal receptor 2) mRNA expression in the recipient cell was observed, making it a viable option for breast cancer therapy. The exogenous loading is more efficient than the direct loading, as it leads to inefficient loading and no effect on the recipient cells [73]. 

Surface engineering of exosomes is significant, as exosomes may be helpful in delivering drugs in vitro, but, for the perfect delivery in vivo, the exosomes need to be engineered. Otherwise, it will be limited due to non-specific toxicity and off-target effects [74]. The exosomes loaded with drugs are engineered by altering the surface with tumour-targeted ligands. Ligands used on the surface are often those ligands that can bind to those overexpressed receptors in the cancer cells [74]. The recipient cells quickly take up the exosomes with antibodies and the ligands through endocytosis, and they can easily escape encapsulation and trapping. The exosome surfaces are often altered using lipophilic linkers to attach monoclonal antibodies [75] Figure 3.

## 5. Exosomes for the Delivery of Chemotherapeutic Drugs

Exosomes can be used as delivery agents to influence signalling pathways, silence genes, and regulate the expression of different proteins. Exosomes perform gene transfection because of their intrinsic role in endogenous gene transfer. The endogenous origin makes exosomes less toxic to the cells and has a minimal immune effect [76]. The human body can tolerate exosomes, as they are naturally present, enabling them to behave as a delivery agent for potent drugs. Doxorubicin loaded in exosomes was developed to target HER2+ tumour cells and showed less cytotoxicity compared with the free delivery of doxorubicin (non-capsulated with exosomes), and, hence, can reduce the side effects caused by the chemotherapeutics drugs [77].

## 6. Engineered Exosomes in Triple-Negative Breast Cancer Drug Delivery

HELA-exosomes (cervical cancer cell lines) are engineered exosomes that exhibit significant anticancer effects in both a mouse model and human breast cancer organoids by increasing the activation of cDC1s in situ and, hence, boosting the tumour-reactive CD8+ T cell responses. Exosomes extracted from mesothelin (MSLN) initiate CAR-T cells, which have abilities similar to T cells, including CARs (cysteinyl-TRNA-synthetase) and CD3 surface expression. The exosomes loaded with CARs are seen to inhibit the growth of both endogenous and exogenous MSLN + TNBC. With the help of proteins such as perforin and granzyme B, exosomes can terminate the tumour. The tumour inhibition rate is high, followed by fewer side effects. Exosomes that have been isolated from CAR-T cells that are mesothelin (MSLN)-targeted significantly hinder the proliferation of MSLN-positive TNBC cells. Exosomes extracted from CAR-T cells exhibit a form of cell-free therapy that is significantly less toxic and more effective at preventing tumour growth than live CAR-T cells [78].

The exosomes derived from the HEK293T (human embryonic kidney cells with adenovirus) cells have LipHA-hEVs (Lipid grafted hyaluronic acid) as a ligand loaded with doxorubicin to target those cells expressing CD44, which often escape from the macrophages in MCF7/ADR cell lines or breast cancer [79]. Therefore, in breast cancer, the tumour mass is reduced by 89 percent, and the animal survival rate is increased by 50 percent [76]. Engineered exosomes with a modified surface coating of poly (lactic-co-glycolic acid) can target C-met, the epithelial–mesenchymal transition factor. Overexpression of C-met is in TNBC cells. The coated exosomes enhance the efficiency of uptake and anti-tumour activity of doxorubicin. In vivo experiments also show that the anti-tumour activity is improved, which induces apoptosis in triple-negative breast cancer [80]. Anti-HER2 antibody-conjugated paclitaxel-loaded liposomes are then employed to deliver HER2-targeted drugs. In vitro and in vivo experiments show that HER2 grafting combined with targeted medication delivery can boost therapy success. This innovative strategy may open the way for the rising wave of targeted medicine and immunotherapy by providing a simple method of modifying cell membranes by antigen presentation via accessible exosome uptake [81].

It is well known that the recently-created chemotherapeutic medication erastin has the ability to cause ferroptosis in TNBC cells, but its use has been limited by its poor water solubility and side effects. To solve this problem, Yu et al. loaded erastin into exosomes labelled with folate (FA) to target the TNBC cells overexpressing folate receptors on the cell surface, thereby enhancing ferroptosis with the pre-depletion of intracellular glutathione and overgeneration of reactive oxygen species (ROS). This significantly reduced the proliferative and migratory abilities of MDA-MB-231 cells [82]. Anti-IL-3R-EV therapy suppresses lung metastasis caused by intravenous injection of MDA-MB-231 cell lines. Anti-IL-3R-EV lowers Vimentin, β-catenin, and TWIST1 expression and practically eliminates liver and lung invasion from primary tumours. Anti-IL-3R-EVs of miR-24-3p (antago-miR-24-3p-EVs) are eliminated in vivo to regulate TWIST1 and diminish proliferative damage. In MDA-MB-231 cells, anti-IL-3R-EVs and anti--24-3p-EVs stimulate the expression of the SPRY2 (Sprouty homolog 2) gene, which is compatible with network analysis of miR-24-3p gene targeting. Finally, SPRY2 knockdown demonstrates that SPRY2 is the target for anti-IL-3R-EVs and induces apoptosis by inhibiting apoptotic stimuli mediated by these molecules and antago-miR-24-3p-EV [83].

One possible option is to increase tissue-specific targeting using tissue-specific antibodies or homing peptides by engineering the innate homing ability of exosomes. A chimeric protein against HER2+ is delivered by engineered exosomes with pLEX-LAMP DARP on their surface [84,85]. Engineered exosomes can target different tissues by masking the exosomes with membrane and surface display with an extra protein. To create a reconfigurable transmembrane adhesion platform, streptavidin is linked to 1,2-bis(dimethylphosphino)ethane: polyethylene glycol 5 k (DMPE-PEG). This enables biotinylated molecules, such as antibodies, to be associated with the anchor and used to construct vesicle surfaces for targeted delivery [86] Tables 1 and 2.

## 7. Exosomes as a Delivery Agent for microRNA and siRNA in TNBC Modulating Signalling

Exosomes generated from adipose mesenchymal stem cells infused with miRNA-381-3p reduce the metastasis in TNBC cell lines. Additionally, it was observed that miRNA-381-3p downregulates the expression of Wnt signalling and elements linked to the epithelial–mesenchymal transition in MDA-MB-231 [87]. miR3182 conjugated with exosomes derived from the human umbilical cord induce apoptosis in the TNBC cell line through downregulating genes such as mTOR and S6KB1 [88]. Exosomal miR-27a-3p, through production by endoplasmic reticulum stress, enhances immunological escape in breast cancer through modulating PD-L1 levels in macrophage’s immune evasion via the PTEN/AKT/PI3K axis [89]. Exosome A15 derived from monocyte-derived macrophages by activating protein kinase C can deliver microRNA 159 and doxorubicin, causing a synergetic effect on the TCF-7 gene and improving the anticancer effects. Exosomes containing miRNA-127, miR-197, miR-222, and miR-223 from stroma also aid breast cancer cell quiescence and enhance anti-proliferation rates, but to a lower extent than miRNAs transferred through gap junctions [90]. The TPD52 gene is downregulated by 70% when engineered exosomes from HEK293T cells are directed to translocate to HER2 positive breast cancer cells [91].

Exosomal miR-335-5p suppresses TNBC immune escape mediated by PD-L1 by promoting the ubiquitination of PD-L1 by USP22 via downregulation of USP22 and PD-L2 [92]. miR-134 is downregulated in the TNBC Hs578Ts(i)8 cell line. miR-134′s expression is lowered, and it acts as the tumour suppressor. It targets the pathways such as MAPK/ERK (mitogen-activated protein kinase/extracellular signal-regulated kinase) and other signallings such as EGFR (epidermal growth factor receptor) and NOTCH [93]. When miRNA-134 is inserted into Hs578Ts(i)8 cells with the help of exosomes, it can downregulate genes such as STAT5B (signal transducer activator of transcription 5) and HSP90. These genes are anti-apoptotic proteins and regulate the targets of miRNA-134. The conjugated exosomes with the microRNA-134 control the migration and the invasion of recipient TNBC cells and enhance recipient cell’s sensitivity to anti-HSP90 inhibitors [94]. Upward expression of miR-134 also represses breast cancer metastasis by targeting KRAS (Kirsten rat sarcoma virus). Therefore, exosomes could be used as a delivery agent and can intervene with the signalling in TNBC [95,96].

The level of microRNA let-7a expression in TNBC cell lines and patients suffering from triple-negative breast cancer shows low expression. The let-7a conjugated with exosomes, when targeted in the TNBC mice model, reduces tumour growth, migration, and invasion. The let-7a binds to the 3’ UTR of the c-Myc gene and silences the gene expression [97]. miR-770, transferred with the help of exosomes, can alter the tumour microenvironment in TNBC. The transmission of miR-770 via exosomes was confirmed in the co-culture of tumour-derived exosomes and tumour-associated macrophages, as revealed by enhanced expression of miR-770 in the recipient cells. Doxorubicin sensitivity is increased in cancer cells by miR-770 and it enhances therapeutically-induced apoptosis [98].

In vivo tumour growth is aided by miR-9. miR-9 is an essential player in the cross-communication between cancer cells and stroma [99]. miR-9 and miR-155 target PTEN (phosphatase and tensin homologue) and DUSP14 (dual specificity phosphatase 14) tumour suppressor genes in MCF-7 cell lines extracted from MDA-MB-231 exosomes [100]. Anti-miR-142-3p can be delivered efficiently by MSCs-Exo, lowering miR-142-3p and miR-150 levels while increasing the expression of the regulatory target genes APC (adenomatous polyposis coli protein regulator of Wnt) and P2X7R (purinergic receptor) in TNBC [101]. miR-496 and miR-137 are loaded in exosomes from MCF10A cell line target Del-1 and modulate it by inhibiting tumour growth in the TNBC cell lines such as MCF10A and MDA-MB-231 [102]. Extracellular vesicles derived from adipose mesenchymal cells enhance apoptosis by suppressing PD-L1 signalling in MDA-MB-231 TNBC cells [103,104] Table 1 and Table 2 and Figure 4.

## 8. Exosomal-Associated Small Molecule Modulating Pathways Inhibiting Metastasis of Triple-Negative Breast Cancer to Other Organs

As per the current report, exosomes delivering lncRNA DARS-AS1 siRNA in TNBC inhibit mild stress induced by TNBC tumours, which causes metastasis [104]. The mesenchymal stem cells derived from exosomal miR-106a-5P are suppressed by LncRNA HAND2-AS1 (link RNA heart and neural crest derivatives expressed 2), influencing the growth of triple-negative breast cancer [105]. Zhao et al. discovered that cationic bovine serum albumin (CBSA) conjugated siS100A4 and exosome membrane-coated nanoparticles are biocompatible and can improve drug delivery to lung PMN (poly morpho nuclear leucocytes). They can silence the genes that inhibit breast cancer’s malignancy in the lungs and are essential for post-overactive breast cancer tumours [106]. The therapeutic index of exosomes loaded with doxorubicin is more than the free one in TNBC. The cardiac toxicity is lowered [107]. miR-4516 can be used as a therapeutic for TNBC, as the CAF-derived exosomes influence the FOSL1 (transcription factor subunit) and initiate TNBC progression in the absence of miR-4516 [108].

## 9. Exosomes Specifically Regulating TNBC Signallings

Endogenous exosomes’ capacity to transmit proteins or miRNA makes them particularly useful for delivering nucleic acid- or peptide-based therapies directly to cancer cells. The siS100A4 is coupled to cationic bovine serum albumin (CBSA). This approach has demonstrated a strong suppressive impact on post-surgery metastasis in TNBC. The expression of the S100A4 protein, which is involved in metastasis, is downregulated by exosome-mediated RNAi, and postoperative metastasis in TNBC is significantly suppressed as a result. These investigations show that an effective therapeutic method for the prevention and treatment of cancer is the administration of siS100A4 through exosome membrane-packed core-shell NPS [109]. (Table 1 and Table 2)

KLF4 (Kruppel-like factor 4) and HOXD4 (homeobox D4) are the transcription factors that play a major role in metastasis and proliferation. MicroRNA miR-10b (microRNA) can transfer in metastatic BC cells (MDA-MB-231) according to different research by Singh et al., 2014 [110]. The target genes are KLF4 (Kruppel-like factor 4) and HOXD10 (homeobox D10). Target genes such as KLF4 and HOXD10 that are connected to protein expression can be silenced by the miR-10b if present in the elevated form [111].

Gong et al. created metalloproteinase-15-rich nanoscale target-specific exosomes that can carry doxorubicin (DOX) and miR-159 that have been modified with cholesterol to TNBC cells. In both in vitro and in vivo scenarios, it has been demonstrated that exosomes co-loaded with DOX and cholesterol-miR159 efficiently inhibit the growth of MDA-MB-231 TNBC cells by downregulating the expression of TCF-7 (transcription factor 7) [112]. The exosome-coated nanoparticles’ precise targeting ability greatly improved cellular absorption efficiency and demonstrated extraordinary TNBC-targeted efficacy, causing cancer cells to die, and, thus, inhibiting the course of the disease. Exosomes made from macrophages and coated in poly (lactic-co-glycolic acid) were highly effective in treating TNBC with targeted chemotherapy [113].

PD-L1 (programmed death ligand 1) is the major biomarker for TNBC [114]. When exosome surface PD-L1 attaches to the anti-PD-L1 antibody, leaving the tumour PD-L1 exposed, or when exosome surface PD-L1 binds to PD-1 on effector T cells during monoclonal antibody therapy, tumour cells can specifically bypass immunosuppression [115]. Therefore, exogenous PD-L1 is required for exosomes to have an immunosuppressive function; this immunosuppression may be prevented by pre-clearing exosomes with an anti-PD-L1 antibody. Furthermore, immune checkpoint blockade (ICB) therapy for metastatic TNBC uses exosomes loaded with PD-L1 as a helpful and reliable biomarker [116].

Exosomes for immunotherapy in TNBC can also be produced by mesenchymal stem cells (MSCs), i.e., MSC-derived exosomes, in addition to lymphocytes. Exosomes made from MSCs play a crucial role in immunomodulation as well, primarily by controlling immune cell activity and modifying the release of inflammatory mediators such as TNF and IL-1 in inflammatory bowel disease [117]. Monocytic myeloid-derived suppressor cells (M-MDSCs) may be differentiated into highly immunosuppressive M2-polarized macrophages with the help of MSC-derived exosomes. Additionally, PD-L1 overexpression in myeloid cells and PD-1 downregulation in T cells in breast cancer in vivo are two additional ways that MSC-derived exosomes might compromise protective anti-tumour immunity [111]. Although several studies have shown exosomes may have a therapeutic use and numerous clinical trials are now using exosomes, no clinically exosome-based medicines have yet received approval. All early research indicates that exosomes are a unique, potential treatment option for TNBC, and there is a high likelihood of employing modified exosomes as multifunctional therapeutic agents for TNBC therapy. Exosomes from the TNBC tumour microenvironment may also be used as a target for vaccination in the future to enhance the treatment effectiveness of TNBC.

## 10. Exosomes from Plants and Bacteria as a Delivery Agent

Derivation of exosomes is also from many other sources such as bacteria and plants. Exosomes from plants and bacteria have shown anti-tumour function. Low toxicity and large production are why exosome usage is attractive in cancer therapy. The tumour microenvironment contains EVs or outer membrane vesicles in the size of 40 nm obtained from gram −ve and +ve bacteria. They can produce anti-tumour cytokines as well as interferon-γ and CXCL10 (C-X-C motif chemokine ligand 10) [118]. Plant-derived extra vesicles are well researched. EVs from grapes, ginger, and citrus fruit act as delivery agents that can deliver chemotherapeutic and small molecular drugs [119]. Exosomes derived from grapefruit loaded with miR-17 and folic acid were delivered by the intranasal route inside mice, which inhibited brain tumours. Hence, we can conclude that breast cancer, when metastasized to the brain, can easily be targeted by using the exosomes, as they can even pass through the blood–brain barrier [120]. More studies in the field of triple-negative breast cancers are yet to be done, and it can be a new therapeutic approach to treat TNBC.

No current therapy involves plant-derived exosomes for the treatment of TNBC. As per the studies after the characterization of exosomes extracted from plants, it can be used as a new therapeutic for TNBC by targeting the pathways such as Wnt signalling, NOTCH, HIPPO signalling, and genes involved in epithelial–mesenchymal transition, which can fulfil the present research gap as mentioned in this review paper. More exploration with plant exosomes is needed.

## 11. Exosome-Mediated Delivery Vehicles in Pre-Clinical and Clinical Trials (Vaccines) in Triple-Negative Breast Cancer

Vaccines against cancer help the human body to redirect the immune system to fight against cancer rather than to prevent it. Standard vaccination leads to the prevention of infection towards other viral or bacterial infections. As a result, it appears that developing preventative EV-based cancer vaccines against virally-produced cancers, such as HPV, and therapeutic EV-based cancer vaccines for non-virally induced cancers, such as TNBC, would be more appropriate. EVs can communicate between the cells and, therefore, can be used as a vaccine. Natural killer cells derived from exosomes have more potential cytotoxicity effects against tumour cells. Exosomes have the potential to be a drug delivery agent in vitro as well as in vivo. Clinically, exosomes are tested as a delivery vehicle in other cancers. Exosome delivery could be a hope for TNBC as well [121,122]. Exosomes loaded with curcumin are tested against colon cancer undergoing clinical phase 1 trials. (NCT01294072).

Exosomes are small vesicles secreted from dendritic cells consisting of MHC class 1 and class II molecules, along with other CD86 proteins and tetraspanins, stimulate an immune response and undergo phase 2 clinical trials [123,124]. The literature already reports that EV cancer vaccines derived from mesenchymal stem cells, macrophages, and tumour-derived exosomes are present. Tumour-derived exosomes can act as the vaccine against cancer. However, consideration of also having oncogenic materials is needed. TEV (tumour-derived extracellular vesicles)-based vaccines contain tumour rejection antigens and can effectively be represented to the dendritic cells by MHC-1 molecules for cancer treatment [125]. As the TEV-based vaccines have safety issues, other alternatives can be used, such as extracellular vesicles from healthy cell lines carrying TAAs (tumour-associated antigens) DNA, protein, or the mRNA [126]. Zitvogel showed that a single dose of a dendritic cell-derived vaccine against cancer could stimulate the immune response via activating T cell-mediated anti-tumour effects [127]. The dendritic cell-derived exosomes can inhibit tumours by activating CTLs (cytotoxic T lymphocytes) and natural killer cells [128]. They have also shown an immune response toward mammary adenocarcinoma tumour cells. DC-EV-based vaccines against cancer can trigger the immune system by presenting them with peptides or TAAs related to cancers [129]. They have also been proven safe and achievable in phase 1 of clinical studies. Advanced research is necessary for the long-lasting response of the immune system against tumour cells in the field of exosome-derived vaccines [130]. As shown in Table 3, [131] six active clinical trials test cancer vaccines for TNBC with chemotherapeutic medicines or immunotherapy treatments [131].

## 12. Conclusions and Perspectives

Exosomes have many promising characteristics—bio-feasibility, biocompatibility, good loading capacity, and less toxicity, being derivatives from biological sources—making them a very appropriate candidate for drug delivery. Exosomes can easily be manipulated accordingly by making changes to their surface with the addition of proteins or antibodies. Many methods are available for loading and also for extraction. Cost-effective production of exosomes is also needed, as there is a limitation with the available methods for the extensive quantity of extractions from the cells. Much scientific data showed how exosomes could reduce the metastatic rate of TNBC by delivering miRNA, siRNA, LncRNA, and chemotherapeutic drugs in breast cancer cell lines. The extraction of exosomes from plants has not been explored yet in the field of TNBC, but scientific proof is present with brain cancer. Hence, the exosomes from plants can also be used as a delivery agent for TNBC. More research is needed in the field to study the characteristics of TNBC and the constituents of TNBC. Exosomes also play a crucial role in the EMT transition in TNBC. Altering the secretions of exosomes by modulating the content or silencing some genes may be a solution for the metastasis of TNBC, which progresses to other organs. Exosomes can also be biomarkers for TNBC that are responsible for chemoresistance in cancers. Exosomes also target signalling such as Wnt, NOTCH, and PTEN/AKT/PI3K (phosphatase tensin homolog/phosphokinase pathway), etc., in the TNBC, which plays a vital role in the epithelial-to-mesenchymal transition. Exosomes can also target genes, making them a perfect target delivery agent. Therefore, a cure for TNBC with personalized therapy is possible if targeted therapy with the help of exosomes is studied more widely. An in-depth study can also pave the way for exosomes to emerge as a vaccine to treat triple-negative breast cancer.

## Figures and Tables

**Figure 1 molecules-28-01802-f001:**
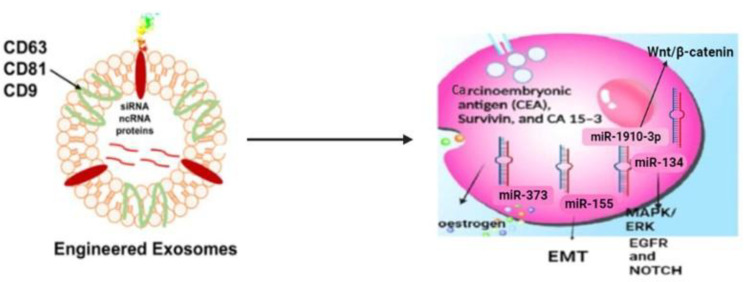
Exosomes as a biomarker for TNBC—showing different protein contents inside exosomes, which are used for identifying TNBC and also in ‘liquid biopsy’. Carcinoembryonic antigen (CEA), survivin, CA 15-3(Cancer antigen 15-3), miR-373, mi-134, miR-1910-3p, and miR-155 are some of the biomarkers (https://biorender.com/, accessed on 30 October 2022).

**Figure 2 molecules-28-01802-f002:**
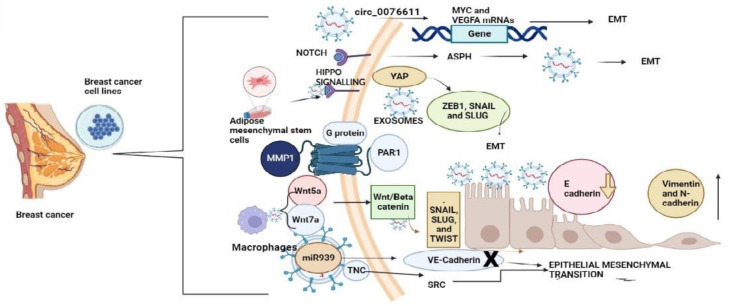
Exosomes targeting different pathways in TNBC and breast cancer leading to epithelial–mesenchymal transition of breast tumours to other parts of the body (exosomes have circ_0076611 target MYC and VEGFA genes and initiate epithelial–mesenchymal transition). NOTCH signalling initiates ASPH in breast cancer cell lines and triggers epithelial–mesenchymal transition. Exosomes extracted from adipose mesenchymal stem cells inhibit Hippo signalling and initiate the TEA domain transcription factor to bind with the unphosphorylated YAP/TAZ, which translocates it to the nucleus. TEAD-influenced target gene transcription enhances the expression of mesenchymal markers such as Vimentin and N-cadherin and suppresses the expression of epithelial markers including E-cadherin. The YAP gene cross-communicates with transcription factors such as ZEB1, SNAIL, and SLUG, which initiates epithelial–mesenchymal transition in cancer cells. MMP-1 interacts with PAR1, a G protein receptor that initiates the EMT. Macrophage-derived exosomes containing Wnt5a and Wnt7a activate Wnt/β signalling, which, in turn, activates SNAIL, SLUG, and TWIST and initiates EMT. miR-939 in exosomes downregulates the expression of VE-cadherin, destroys the barrier function of endothelial monolayers, and initiates tumourigenesis in breast cancer. (Created with the help of Biorender).

**Figure 3 molecules-28-01802-f003:**
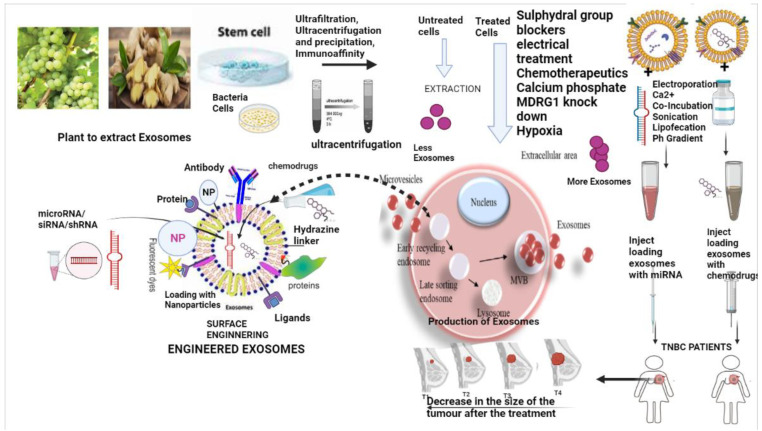
Extraction of exosomes from different sources such as ginger and grapes, stem cells, and bacterial cells using different methods such as ultracentrifugation; the influence of external agents on the production of exosomes; the loading of the exosomes with miRNA, siRNA, and drugs; engineering of exosomes by protein, antibody, nanoparticle, or hydrazine linker, finally using it as drug delivery; and its effect on the tumour (https://biorender.com/, accessed on 30 October 2022).

**Figure 4 molecules-28-01802-f004:**
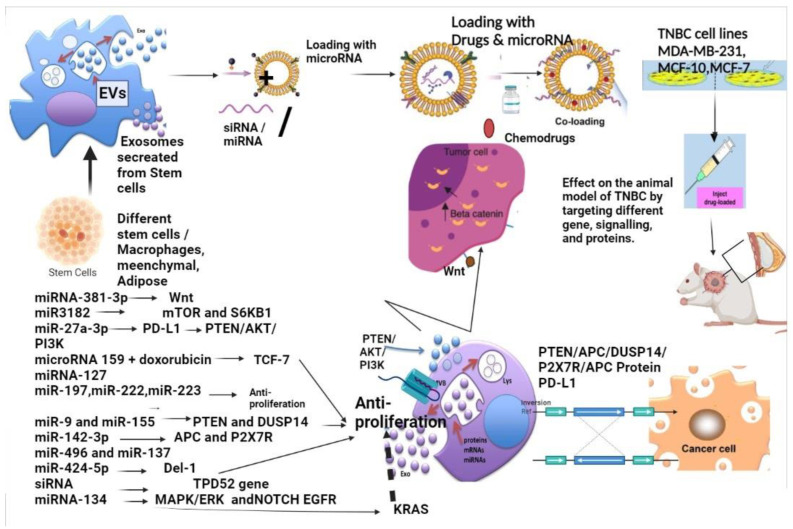
Exosomes generated from adipose mesenchymal stem cells infused with miRNA-381-3p downregulate the expression of Wnt signalling and inhibit EMT. miR3182 conjugated with exosomes derived from the human umbilical cord induce apoptosis in the TNBC cell line through downregulating genes such as mTOR and S6KB1. Exosomal miR-27a-3p, through production by endoplasmic reticulum stress, enhances immunological escape in breast cancer through modulating PD-L1 levels in macrophage’s immune evasion via the PTEN/AKT/PI3K axis. microRNA 159 and doxorubicin cause a synergetic effect on the TCF-7 gene and improve the anticancer effects. Exosomes containing miRNA-127, miR-197, miR-222, and miR-223 from stroma also aid breast cancer cell quiescence and enhance anti-proliferation rates. miR-134 targets the pathways such as MAPK/ERK and other signallings such as EGFR and NOTCH. Upward expression of miR-134 also represses breast cancer metastasis by targeting KRAS. miR-9 and miR-155 target PTEN and DUSP14 tumour suppressor genes in MCF-7 cell lines extracted from MDA-MB-231 exosomes. miR-496 and miR-137 loaded in exosomes from the MCF10A cell line target Del-1 and modulate it by inhibiting tumour growth in the TNBC cell lines such as MCF10A and MDA-MB-231. Extracellular vesicles derived from adipose mesenchymal cells enhance apoptosis by suppressing PD-L1 signalling in MDA-MB-231 TNBC cells.

**Table 1 molecules-28-01802-t001:** Engineered exosomes with different proteins embedded their membranes, and their effects on triple-negative breast cancer.

Engineered Exosomes Element in Membrane/Source	Effect	Reported Effect and Outcomes	References
pLEX-LAMP DARPin for expression of DARPin	HER2+ cells	Inhibit metastasis	[84,85]
Mesothelin MSLN	CAR-T cells with CARs and CD3 surface expression	Inhibit metastasis	[78]
Poly (lactic-co-glycolic acid)	MDA-MB-231	Target c-Met and inhibit metastasis	[80]
Anti-HER2 antibody conjugated paclitaxel-loaded liposomes	MDA-MB-231	Inhibit metastasis, boost the therapy, increase apoptosis	[81]
Exosome A15 derived from monocyte-derived macrophages	MDA-MB-231	Effect on TCF-7 gene leading to improved anticancer effects	[90]

**Table 2 molecules-28-01802-t002:** Engineered exosomes loaded with different molecular drugs, microRNA, siRNA targeting different signalling, and genes in triple-negative breast cancer.

Cargo	Effect	Reported Effect and Outcomes	Refs.
miR-134	Hs578Ts(i)_8_	Downregulate STAT5B, HSP90, and KRAS	[93,94,95,96]
let-7a	MDA-MB-231	Silence c-Myc gene	[97]
miRNA-770 and Doxorubicin	MDA-MB-231	HER2+ tumour	[98]
CAR	MSLN + TNBC	Perforin and Granzyme B mechanism	[78]
Erastin	MDA-MB-231	Ferroptosis	[82]
Anti-IL-3R-EV	Metastasis of TNBC to other parts, liver, lungs	Lower Vimentin, β-catenin, and TWIST1	[83]
Anti-IL-3R-EVs and antago-miR-24-3p-EVs	MDA-MB-231	Upregulate SPRY2, enhance apoptosis	[83]
miRNA-381-3p from adipose mesenchymal stem cells	MDA-MB-231	Decrease Wnt signalling and factors related to EMT	[87]
miR3182	MDA-MB-231	Apoptosis in TNBC to downregulate mTOR and S6KB1	[88]
miR-27a-3p	MDA-MB-231	Modulate PD-L1 levels in macrophages causing immune evasion via the PTEN/AKT/PI3K axis	[89]
microRNA 159	MDA-MB-231	Activate protein kinase C further TCF-7	[90]
miRNA-127, miR-197, miR-222, and miR-223	MDA-MB-231	Enhance apoptosis, metastasis	[90]
siRNA from HEK293T cells	HER2 Positive Breast Cancer	TPD52 gene downregulated by 70 percent	[91]
miR-9 and miR-155 from MDA-MB-231	MCF-7	*PTEN* and *DUSP14* tumour suppressor gene	[100]
Anti-miR-142-3p	MDA-MB-231	Lower miR-142-3p and miR-150 levels while increasing the level of target genes APC and P2X7R in TNBC	[101]
miR-496 and miR-137 from MCF10A	MDA-MB-231	Del-1	[102]
miR-424-5p	MDA-MB-231	Suppress PD-L1 enhancing apoptosis	[103]

**Table 3 molecules-28-01802-t003:** List of all clinical trials with exosomes loaded with drugs in triple-negative breast cancer [131].

Trial (National Clinical Trial ID)	Phase	Condition	Interventions
NCT03362060	1	TNBC, Metastatic TNBC	Pembrolizumab Biological: PVX-410
NCT04105582	1	TNBC BC	Biological: Neo-antigen pulsed DCs
NCT03199040	1	TNBC Metastatic TNBC	Drug: Durvalumab Biological: Neoantigen DNA vaccine Device: TDS-IM system (Anchor Medical Systems)
NCT02316457	I	TNBC	Biological: IVAC_W_bre1_uID Biological: IVAC_W_bre1_uID/IVAC_M_uID
NCT04024800	II	TNBC	Biological: AE37 Peptide vaccine
NCT02826434	1	BC	Biological: PVX-410 Biological: Durvalumab Drug: Hilton

## Data Availability

Not applicable.

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
