# Peer review of "Critical Review on the Different Roles of Exosomes in TNBC and Exosomal-Mediated Delivery of microRNA/siRNA/lncRNA and Drug Targeting Signalling Pathways in Triple-Negative Breast Cancer"

_molecules, 2023, doi:10.3390/molecules28041802_

Round 1

Reviewer 1 Report

The manuscript from Banerjee and Rajeswari compiled background information that exosomes might be a new therapeutic approach for TNBC treatment.

After a short overview about the characteristics and problems in the treatment of triple negative breast cancer, the manuscript contains information to exosomes in general and exosomes in TNBC (biomarker, diagnosis, EMT). The following chapters address exosome production, exosomes as delivery agents for drugs, RNAs and small molecules and at the end their use in TNBC treatment in pre-clinical studies.

Specific comments:

Page 2 line 63: “To fulfil these demands, the scientist devised the extracellular vesicles, i.e., exosomes, as the natural delivery system…..” à exosomes used for the delivery of chemo-therapeutics are not “natural”. This term is misleading in this context.

Figure 1: the figure is not well structured. Is the right picture of Figure 1 an exosome? If so, it should not contain a nucleus. Furthermore CEA the abbreviation for “carcinoembryonic antigen”, please correct in the figure.

Page 4 line 159: “Zinc finger protein” is mentioned twice.

Line 161/162:” The exosomes are involved in intercommunication between the cells.”à between which cells? Please specify.

Page 4/5 line 176 and 181: “YAP gene cross-communicate with transcription factors like ZEB1, SNAIL and SLUG which initiates epithelial-mesenchymal transition in cancer cells.”and “SNAIL, SLUG, and TWIST are the transcription factors that get initiated and trigger EMT” are very similar sentences, please revise.

Figure 2: The figure is confusing. Why are the different breast cancer stages included at the left side? What is the left and right side of the membrane? Which pathways are targeted by exosomes of which cells (macrophages, adipose mesenchymal stem cells…)?  The legend to this figure should contain a clear description of the figure and a detailed explanation of the symbols.

Figure 3: Also this figure is confusing and a detailed description in the legend is missing.

In general, the review does not clearly disclose how exosomes can be specifically target TNBC cells. Even there are several pathways depicted that might be good targets (but also here the specificity for TNBC is not always clear), the delivery of exosomes to a specific cell type depends on specific marker molecules, which is the main problem in TNBC and other cancer. Mice studies could not be used to predict the outcome in humans and several off target effects (often due to unspecific side effects) are only obvious when the drugs are tested in vivo. Therefore, the title of this review “A new therapeutic approach for TNBC involving engineered exosomes for the delivery of chemotherapeutics drugs and microRNA/siRNA/lncRNA targeting signalling pathways” is misleading, because most data presented here rely on in vitro studies, using TNBC cells in cell culture and are far away to be used as new a therapeutic approach for TNBC treatment.

Several abbreviations are not explained in the text (e.g. HELA-exosomes, MSLN, CAR, LipHA-hEVs, ADR, TEV, TAA’s DNA…..)

English has to be revised: e.g. Page 8 line 300 “The tumour inhibition rate was very much high, with fewer side effects.

Reviewer 2 Report

The field of exosomes is continuously growing and hundreds of publication present new approaches for analysis or for medical applications. Also, many reviews in this growing area of study are issued every year.

Unfortunately, the MS “A new therapeutic approach for TNBC involving engineered 2 exosomes for the delivery of chemotherapeutics drugs and mi-3 croRNA/siRNA/lncRNA targeting signalling pathways” do not face this competition and fails to bring an interesting new point of view.

Firstly, the English is very poor. Although is good to have simple phrases, the authors put the whole manuscript in a simple key that makes it very unattractive. The chapters and sub-chapters have a poor structure and the continuity specific for a good review is totally missing.

The titles of the MS and the subtitles are difficult to understand and sometimes they are misleading. For instance, "Exosomes as a signature molecule in TNBC" is wrong firstly by the definition of exosomes as "molecules" and secondly, from the content, as the text describes specific cancer proteins without arguments for exosome isolated proteins.

The pictures are not visible and have poor captions. Altogether the paper is difficult to read and moreover does not “describe a new therapeutic approach” as promised in the title. I recommend the rejection of this manuscript.

Round 2

Reviewer 1 Report

The authors substantially improved this review in the revised version.